# Beyond coverage: Can vaccination reduce child mortality despite structural inequality?
A global ecological analysis of routine childhood immunization (2010–2023)

**Valeria Rios Montoya**[1], **Arlet Yelisse Suarez Alarcon**[1], **Shamira Perez Castillo**[2], **Trilce Zamudio Raez**[1], **Antonio Marty Quispe**[3,4]*

**1** Departamento de Bioingeniería e Ingeniería Química; Universidad de Ingeniería y Tecnología, Lima, Peru, **2** Departamento de Ingeniería Civil y Ambiental, Universidad de Ingeniería y Tecnología, Lima, Peru, **3** Escuela de Posgrado, Universidad Señor de Sipán, Chiclayo, Peru, **4** Centro de Investigación en Bioingeniería, Universidad de Ingeniería y Tecnología, Lima, Peru

* drantonioquispe@gmail.com

## Abstract

Routine childhood immunization is still a critical topic to global public health, since it prevents an estimated 3.5 to 5 million deaths annually. However, its access remains uneven, especially in low-income countries, where structural inequalities limit the reach of immunization programs, and the Covid-19 pandemic disrupted routine services, worsening the existing disparities. This ecological study (2010–2023) examined associations between vaccine coverage and infant mortality using the WHO/UNICEF data for six childhood vaccines: BCG, DTP3, HepB3, Hib3, MCV2, and Pol3. Countries were stratified by income level using the World Bank World Development Indicators. Associations between vaccine coverage and infant mortality were evaluated using multivariable linear regression models adjusted for national income level. In addition, non-linear relationships and rank-based associations were explored using LOESS smoothing and Spearman correlation analyses. Results showed higher coverage and lower mortality in high-income countries; meanwhile, low-income countries faced both reduced coverage and higher mortality rates. A significant decline in coverage occurred in 2020, with only partial recovery by 2023. After adjusting for income, most vaccine coverage indicators lost statistical significance in relation to infant mortality. These findings highlight that income-related structural inequities determine immunization coverage and preventable child mortality, emphasizing the need for policies that simultaneously expand vaccine access, reduce structural barriers, and strengthen health systems.

**Data availability statement:** The data supporting this study's findings are open data curated by the World Health Organization (WHO) and the United Nations Children's Fund (UNICEF), which are available under the Creative Commons Attribution 4.0 (CC BY 4.0) license. The data and code are freely available at https://doi.org/10.6084/m9.figshare.30958886.

**Funding:** The authors received no external funding for this study. Universidad Señor de Sipán provided institutional salary support to Dr. Antonio M. Quispe as part of his regular academic appointment. The institution had no role in study design, data collection and analysis, decision to publish, or preparation of the manuscript.

**Competing interests:** The Universidad Señor de Sipan provided support in the form of salary for Dr. Antonio M Quispe, but did not have any additional role in the study design, data collection and analysis, decision to publish, or preparation of the manuscript. This does not alter our adherence to PLOS ONE policies on sharing data and materials. The remaining authors declare no conflict of interest. No external funding was received for this study.

## Introduction

Routine childhood immunization remains as one of the most effective public-health interventions for its impact in reducing under-five mortality worldwide. A recent global analysis estimated that routine vaccination has prevented millions of child deaths across more than 200 countries between 1990 and 2019 [1]. However, several studies have registered substantial inequalities in vaccine coverage across and within low–and middle–income countries, which reflects the structural socioeconomic barriers to secure equitable access [2–3]. These inequities limit the real impact of vaccination programs and affect the most vulnerable populations by undermining the potential of immunization to close gaps in child survival [4–6]. Due to the stagnation– or even reversal in immunization coverage caused by disruptions such as the COVID- 19 pandemic– it is critical to systematically examine how global inequities in vaccine coverage can translate into avoidable child mortality across countries with different income levels [7–9].

Despite the progress, to date, few global studies have systematically examined the socioeconomic gradients across multiple routine childhood vaccines and their association with infant mortality over a prolonged period (2010–2023) [10]. As mentioned above, previous studies have identified socioeconomic, gender, education and geographic access inequalities in vaccination coverage within low-and middle-income countries [11–13]. However, recent reviews such as the one from WHO/UNICEF on immunization inequalities better illustrate the picture by documenting that a large percentage of unvaccinated children are concentrated in a handful of countries; however, those studies have not yet analyzed the impact of those inequalities on preventable child mortality [14]. On the other hand, even though multiple studies have demonstrated the decline in several antigens' coverage due to the COVID-19 pandemic [15–17], there is no consolidated evidence on how such disruptions have altered the balance of structural inequities and their impact on infant mortality globally [18–20]. As a consequence, the key question remains: to what extent do the structural inequalities limit the real and equitable benefit of childhood vaccination on child mortality? Addressing this gap is essential in order to inform global policies designed to maximize the population-level impact of vaccination through an equity-oriented approach.

With this framework, the present study aims to quantify the association between routine child immunization coverage and child mortality across countries at different income classifications between 2010 and 2023, while accounting for country-level socioeconomic conditions. Adopting a global ecological approach, we analyze for six routine childhood vaccines (BCG, DTP3, HepB3, Hib3, MCV2 and Pol3) the temporal trends in coverage, examine income-related gradients in child mortality and determine if the vaccination coverage is independently associated with mortality reductions after adjustment for structural socioeconomic factors. By jointly examining vaccination coverage and mortality outcomes over and extended time horizon, this study aims to clarify the extent to which routine vaccination may mitigate income-related structural inequalities. The results are intended to support evidence-based policy decisions and emphasize the importance of strengthening

health systems, addressing structural barriers, and increasing investment in immunization programs to achieve sustained improvements in child survival.

## Materials and methods

### Ethical statement

This study relied exclusively on publicly available, aggregated country level data retrieved from international open access sources, including WHO, UNICEF, and the World Bank. All data were fully de-identified and contained no personal or confidential information, and no human participants were directly involved in the study. Therefore, this research was exempt from institutional ethics review in accordance with international guidelines for studies using non-identifiable secondary data.

### Study setting

This study was conducted at global level using country-year data from 2010 to 2023. The analytic unit was the country, considering low-, lower-middle-, upper-middle-, and high-income countries as classified by the World Bank during the study period. Data on routine childhood vaccination coverage for key antigens (BCG, DTP3, HepB3, Hib3, MCV2 and Pol3), infant mortality rates and macroeconomic indicators were obtained from publicly available international databases compiled by WHO, UNICEF, and the World Bank. These sources provide comparable estimates across countries and over time, which enables the analysis of income-based gradients in immunization coverage and child health outcomes at a global scale.

### Study design

We conducted a global ecological study to examine the association between routine childhood vaccination coverage and infant mortality across countries with different income levels between 2010 and 2023. Descriptive analyses were first performed to characterize vaccination coverage, infant mortality rates and gross domestic product (GDP) per capita by income group. Given the non-normal distribution of the data, income-related differences were analyzed using non-parametric tests. Multivariable linear regression models were fitted to evaluate the association between vaccination coverage and infant mortality, controlling for income group, with high-income countries designated as the reference category. Additional exploratory analyses included locally estimated scatterplot smoothing (LOESS) to explore potential non-linear trends, as well as Spearman's rank correlation coefficients to assess monotonic associations across country–year observations.

### Study population

The study population includes all countries with available country-level data on routine childhood vaccination coverage, infant mortality, and socioeconomic indicators between 2010 and 2023.Countries were classified according to the World Bank into low-, lower-middle-, upper middle-, and high-income groups income classifications for the corresponding years. To ensure data quality and comparability, the integrity of country-year observations was evaluated across all variables; the number of missing values (NAs) per country ranged from 84 to 540 across the study period. Countries with more than one-third of the maximum observed number of missing values (>180 NAs) were excluded from the analysis (to access the complete country-year analytical dataset, please review the S3 Table, and S4 Table in S1 File for its descriptive characteristics). This threshold was applied to minimize bias from excessive missing data while the final analytical sample was restricted to countries meeting this inclusion criterion. Therefore, the resulting study population included countries with sufficient longitudinal information for key childhood vaccine antigens (BCG, DTP3, HepB3, Hib3, MCV2 and Pol3), infant mortality rates and gross domestic product (GDP) per capita. Descriptive characteristics of

vaccination coverage, infant mortality and GDP per capita across income groups in the included countries are summarized in Table 1.

## Data sources and variables

Country-level data on routine childhood vaccination coverage were obtained from publicly available international databases compiled by the World Health Organization (WHO) and the United Nations Children's Fund (UNICEF), while infant mortality and socioeconomic indicators were obtained from the World Bank's database, all of them for the period 2010–2023. Vaccination coverage indicators included the proportion of children receiving Bacillus Calmette-Guérin (BCG), which protects against severe forms of childhood tuberculosis; the second dose of measles-containing vaccine (MCV2) targets measles infection and is typically administered during the second year of life (around 15–18 months); a later schedule is associated with an increased risk of missed follow-up visits and drop-out between measles vaccine doses in routine immunization programs [21–22]; and the third dose of diphtheria-tetanus-pertussis (DTP3), hepatitis B (HepB3), *Haemophilus influenzae* type b (Hib3), and poliovirus (Pol3) vaccines, which prevent major infectious diseases as diphtheria, tetanus, pertussis, hepatitis B infection, invasive *Haemophilus influenzae* type b disease and poliomyelitis respectively. These antigens are the core components of routine child immunization programs and have contributed substantially to global reduction in vaccine-preventable child mortality. Despite this effectiveness, vaccination coverage remains uneven across countries, reflecting structural inequalities, including socioeconomic disadvantage, geographical barriers to healthcare access, and variations in health system capacity, particularly in low-income settings.

Infant mortality—also referred to as under-one mortality —was defined as the number of deaths among children under one year of age per 1,000 live births, as reported in the World Bank World Development Indicators. Socioeconomic status was defined using the national income classification and gross domestic product (GDP) per capita (current US dollars), which were included as structural covariates to capture income-related differences across countries. All variables were standardized at the country-year level to ensure comparability across data sources and over time.

## Study outcomes

The primary outcome of this study was infant mortality, defined as the number of deaths among children under one year of age per 1,000 live births at the country-year level. Other key analytical outcomes included country-level coverage of

**Table 1. Vaccination coverage, infant mortality and GDP per capita by income level.**

| Income level | VCI DTP3 | VCI Measles (MCV2) | VCI BCG | VCI Polio (Pol3) | VCI Hepatitis B (HepB3) | VCI Hib (Hib3) | Infant mortality | GDP per capita (USD) |
|---|---|---|---|---|---|---|---|---|
| Low income | 92 (91-96) | 71 (61-83) | 97 (92-98) | 91 (89-95) | 92 (91-96) | 92 (90-96) | 38.7 (33.7-46.7) | 731 (587-880) |
| Low- middle income | 93 (86-98) | 93 (74-97) | 96 (92-99) | 94 (86-98) | 94 (86-98) | 93 (85-98) | 17.2 (13.9-23.9) | 3,284 (2,199–3,853) |
| Upper-middle income | 94 (86-97) | 88 (76-96) | 97 (93-99) | 94 (86-97) | 94 (86-97) | 92 (85-97) | 12.6 (8.0-16.7) | 7,560 (6,050−10,032) |
| High income | 96 (93-98) | 91 (86-95) | 98 (95-99) | 95 (93-98) | 95 (91-97) | 95 (92-98) | 3.5 (2.7-5.9) | 34,019 (20,758−51,912) |
| p-value (K-W) | <0.001 | <0.001 | 0.006 | <0.001 | <0.001 | <0.001 | <0.001 | <0.001 |

*Note:* Values were expressed as median (Q1-Q3). Infant mortality is expressed as the number of deaths of infants under one year of age per 1,000 live births. The bottom row shows p-values from Kruskal-Wallis's test. VCI, vaccine coverage index; GDP, gross domestic product; USD, United States Dollars; DTP3, third dose of diphtheria–tetanus–pertussis–containing vaccine; MCV2, second dose of measles-containing vaccine; BCG, Bacillus Calmette–Guérin vaccine; Pol3, third dose of poliovirus-containing vaccine; HepB3, third dose of hepatitis B vaccine; Hib3, third dose of Haemophilus influenzae type b vaccine.

routine childhood vaccines (BCG, DTP3, HepB3, Hib3, MCV2 and Pol3), which were examined to characterize income-related gradients in immunization coverage. In addition, the study examined income-stratified differences in both vaccination coverage and infant mortality before and after adjustment for socioeconomic factors, in order to quantify the extent to which structural inequalities can shape preventable child mortality.

## Statistical analysis

Several descriptive analyses were conducted to summarize vaccination coverage, infant mortality and socioeconomic indicators across countries and income groups over the selected study period (2010−2023). Temporal trends in vaccination coverage were examined graphically to recognize changes over time, including disruptions associated with the COVID- 19 pandemic. Associations between routine childhood vaccination coverage and infant mortality were first explored using non-parametric methods. Spearman correlation coefficients were estimated to analyse rank-based associations between vaccine coverage and infant mortality across country-year observations, and locally estimated scatterplot smoothing (LOESS) was applied to visualize potential non-lineal relationships.

In order to quantify the association between vaccination coverage and infant mortality while taking into account socioeconomic differences, multivariable linear regression models were fitted at the country-year level. Infant mortality was specified as the dependent variable, and vaccination coverage indicators were included as the key explanatory variables. Countries were stratified by the World Bank income group, with high-income countries specified as the reference category. Models were adjusted for national income level and gross domestic product (GDP) per capita to account for structural socioeconomic confounding. Regression analyses were conducted both with and without income adjustment to examine the extent to which observed associations between vaccination coverage and infant mortality were explained by underlying socioeconomic conditions. In addition, because routine immunization services were widely disrupted during the COVID-19 pandemic, the study period (2010–2023) was intentionally retained to capture both pre-pandemic trends and pandemic-related deviations in vaccine coverage. Rather than excluding these observations, country-year data from 2020 to 2023 were maintained to evaluate changes in coverage trajectories. Temporal trends were, therefore, interpreted considering this global disruption, which allowed the analysis to capture both the abrupt decline in vaccination coverage in 2020 and the subsequent partial recovery during 2021–2023. All statistical analyses were conducted using R version 2024.04.2. Model assumptions were evaluated through standard diagnostic procedures, including inspection of residual distributions and assessment of multicollinearity among predictors. Additional details, supplementary tables and figures supporting the analyses are provided in the *Supporting Information* (S1 Table, S2 Table and S6 Table in S1 File).

## Results

A total of 118 countries meeting the predefined data integrity criteria were included in the analysis across the period 2010–2023. Results are presented sequentially to describe temporal trends in routine childhood vaccination coverage, followed by analyses of the association between vaccination coverage, national income level and infant mortality.

Temporal trends in global routine childhood vaccination coverage between 2010 and 2023 were first examined, to characterize pre-pandemic patterns, disruptions during the COVID-19 period and subsequent recovery. Fig 1 shows that, prior to 2020, vaccination coverage was generally stable or showed modest increases—or decreases— across antigens, with BCG and DTP3 consistently exhibiting the highest coverage levels, while MCV2 remained lower throughout the study period. In 2020, a pronounced decline in coverage was observed across all vaccines, and later it increased again from 2021 onwards; however, by 2023, coverage level for all antigens remained below those observed before the pandemic.

Given these global patterns over time, vaccination coverage and infant mortality were compared across national income groups. Table 1 summarizes vaccine coverage, infant mortality, and GDP per capita by World Bank income classification. Clear gradients were observed across all indicators. High-income countries consistently exhibited the highest median coverage for all vaccines, generally exceeding 90%, alongside the lowest infant mortality rates. In contrast,

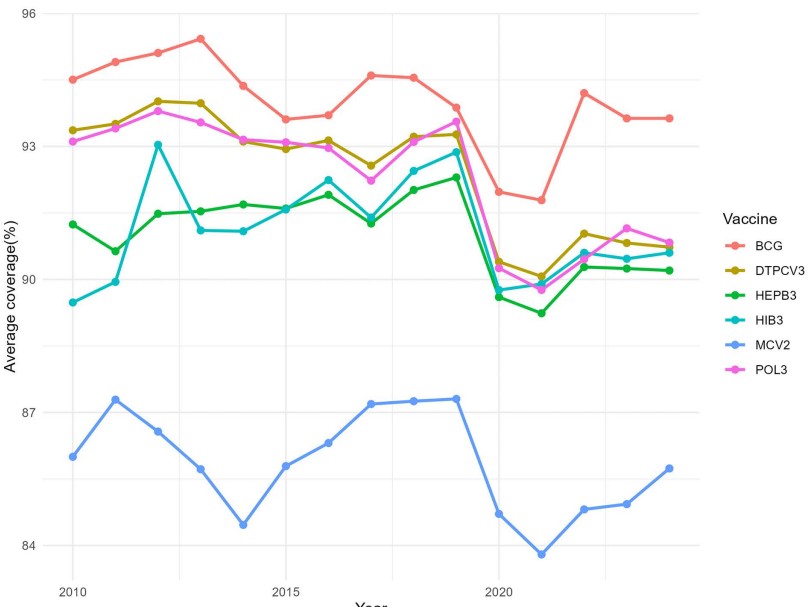

**Fig 1. Global trends in average coverage of routine childhood vaccines, 2010-2023.** Lines represent annual mean coverage (%) for BCG, DTP3 (referred as DTPCV3), HepB3, Hib3, MCV2 and Pol3 across all countries included in the analysis.

low-income countries showed lower vaccination coverage and significantly higher infant mortality. Lower-middle- and upper-middle-income countries displayed intermediate patterns. Differences across income groups were statistically significant for all vaccination indicators, infant mortality and GDP per capita (Kruskal-Wallis test, all p < 0.001), with the exception of BCG vaccination coverage (p = 0.006). These results indicate notorious socioeconomic disparities in both immunization coverage and child survival.

To further examine whether differences in vaccination coverage were independently associated with infant mortality after taking into account the socioeconomic context of each country, multivariable linear regression models were fitted (Table 2). Using high-income countries as the reference category, the national income level emerged as the strongest predictor of infant mortality. If it is compared with high-income countries, infant mortality was significantly higher in upper-middle-income ($\beta = 6.33$; 95% CI: 4.91 to 7.74), lower-middle-income ($\beta = 12.11$; 95% CI: 10.55 to 13.67) and low-income countries ($\beta = 26.94$; 95% CI: 24.49 to 29.40), with all associations being statistically significant (p < 0.001).

After adjusting for income level, most vaccination coverage indicators were not independently associated with infant mortality: coverage of MCV2 showed a modest inverse association with infant mortality ($\beta = -0.28$; 95% CI: −0.33 to −0.24), whereas coverage of the other antigens were not significantly associated with infant mortality (S7 Table in S1 File).

To complement the regressions findings and explore potential non-linear relationships, non-parametric analyses were conducted. Fig 2 represents the association between infant mortality and coverage of six routine childhood vaccines across all country-year observations, stratified by income level. Spearman correlation analyses demonstrated statistically significant inverse associations between vaccination coverage and infant mortality for all antigens ($\rho$ ranging from −0.10 to −0.27; all p < 0.001). LOESS smoothing revealed non-linear patterns, characterized by notorious declines in infant mortality at intermediate levels of coverage and flatter gradients at higher coverage levels. Observations with higher infant mortality were predominantly concentrated among countries with lower vaccination coverage and lower income levels.

Finally, to visually integrate vaccination coverage with national income level, Fig 3 shows the distribution of countries according to coverage and income group for the six vaccines evaluated. Across all antigens, countries classified as high

**Table 2. Regression analysis of infant mortality by income level and vaccination coverage.**

| Associated factors | β(CI 95%) | p-value |
|---|---|---|
| Reference: High income | | |
| Upper-middle income | 6.33 (4.91–7.74) | <0.001 |
| Lower-middle income | 12.11 (10.55–13.67) | <0.001 |
| Low income | 26.94 (24.49–29.40) | <0.001 |
| DTP3 Coverage (DTPCV3) | −0.01 (−0.19–0.18) | 0.953 |
| Measles Coverage (MCV2) | −0.28 (−0.33—0.24) | <0.001 |
| BCG Coverage | 0.05 (−0.01–0.12) | 0.091 |
| Polio Coverage (Pol3) | 0.01 (−0.16–0.18) | 0.889 |
| Hep B Coverage (HepB3) | 0.03 (−0.08–0.15) | 0.564 |
| Hib Coverage (Hib3) | 0.04 (−0.00–0.09) | 0.074 |

*Note:* Multivariable linear regression models were estimated using country-year observations with infant mortality (deaths per 1,000 live births) as the dependent variable. High-income countries served as the reference category. β coefficients indicate absolute changes in infant mortality.

income were disproportionately concentrated in the high-coverage category, whereas low-income countries were more frequently represented in medium- and low-coverage groups. This pattern was consistent in vaccines with traditionally high global coverage, such as BCG and DTPCV3. Notably, MCV2 showed the greatest dispersion across coverage categories, with a substantial proportion of low- and lower-middle-income countries remaining in the low-coverage category. Overall, the figure shows a clear income-related gradient in routine childhood vaccination coverage across antigens.

Taken together, these findings show that higher routine childhood vaccination coverage is generally related with lower infant mortality, while meaningful differences persist across different income groups. The attenuation of vaccination-mortality associations after being adjusted for income, together with the observed income-related clustering of vaccination coverage, indicates that socioeconomic context influences a large part of the variability in the infant mortality rate observed across countries.

## Discussion

This global ecological analysis shows persistent socioeconomic inequalities in routine childhood vaccination coverage and infant mortality from 2010 to 2023. The study identified three principal findings: (i) meaningful income-related gradients persisted in vaccination coverage across all antigens (with low-income countries consistently displaying lower coverage than high-income countries), (ii) higher vaccination coverage was associated with lower infant mortality at the ecological level; however, the association was weakened after being adjusted for national income, highlighting the dominant role of socioeconomic context, and (iii) vaccination coverage declined sharply in 2020 across all antigens, with only partial recovery by 2023. Together, these findings reveal the coexistence of immunization progress and persistent structural inequities in child survival.

The income gradients observed in routine childhood vaccination coverage are consistent with prior evidence that documented socioeconomic inequalities in immunization. Studies in low- and middle-income countries repeatedly demonstrated that full immunization coverage depends strongly on household wealth, maternal education and geographic access to health services [3,6,7]. Likewise, global studies further revealed that children from disadvantaged socioeconomic backgrounds are disproportionately represented among those who are unvaccinated or under-vaccinated [2,9,11]. Even though some regional analyses suggest that expanding the coverage can reduce some inequalities over time, the gaps between income groups often persist [5]. The present study contributes to this literature by demonstrating that these

## Infant mortality vs. vaccine coverage
Colors indicate income level (World Bank classification)

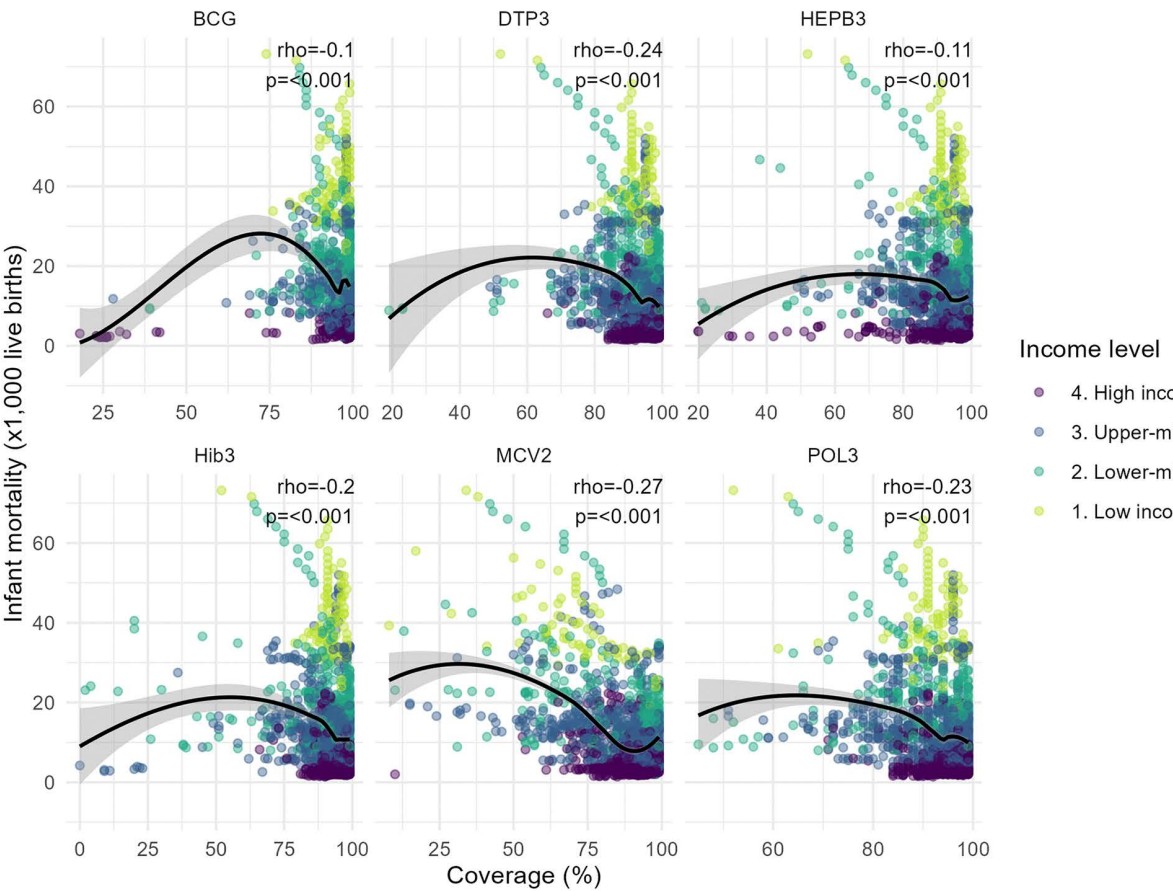

**Fig 2. Association between routine childhood vaccination coverage and infant mortality at the global level, 2010–2023.** Each point represents a country-year observation. The black line represents locally estimated scatterplot smoothing (LOESS). Spearman's rank correlation coefficient ($\rho$) and p-values quantify the strength and statistical significance of the association between vaccination coverage and infant mortality. Points are colored by World Bank income classification.

income-related disparities are not limited to specific regions or vaccines, but rather represent a global pattern across multiple routine antigens over more than a decade.

A second key finding was that, although higher vaccination coverage was associated with lower infant mortality, outliers were still found. After adjusting for income level, the association weakened, which is consistent with prior modeling studies that showed the contribution of vaccination to reduce child mortality, while also emphasizing that its impact operates within broader social and economic contexts [1,4,8]. Structural determinants such as nutrition, maternal education, and access to basic services simultaneously influence both immunization coverage and child survival [11,12,20]. As a result, vaccination alone may be insufficient to counteract the effects of socioeconomic disadvantages. The attenuation observed in adjusted models suggests that income-related structural factors are responsible for much of the observed relationship between coverage and infant mortality at the country level.

A third major finding of this study was the abrupt decline in routine childhood vaccination coverage observed in 2020 across all antigens, followed by only partial recovery by 2022−2023. This pattern is consistent with global

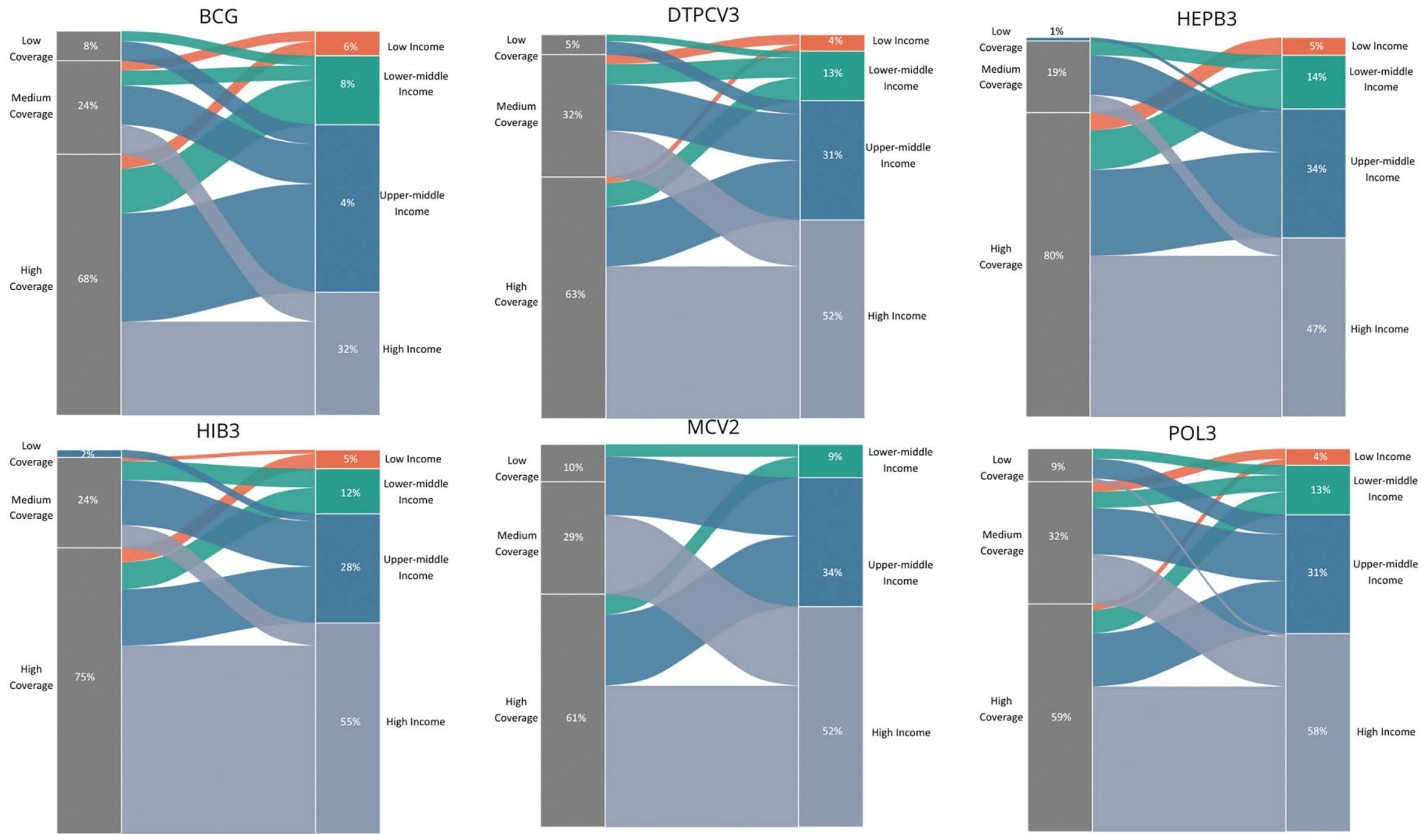

**Fig 3. Distribution of routine childhood vaccination coverage by national income level across six antigens.** Alluvial plots illustrate the distribution of countries according to vaccination coverage categories (low, medium and high) and World Bank income classification (low, lower-middle, upper-middle and high income) for six routine childhood vaccines (BCG, DTPCV3, HepB3, Hib3, MCV2 and Pol3) (S5a-S5d Tables, S8a-S8d Tables and S1-S6 Figs in S1 File). The width of each category represents the proportion of countries within each coverage-income combination.

evidence that document the widespread disruptions to vaccination services during the first year of the COVID-19 pandemic, also driven by health system reallocation, movement restrictions, supply-chain interruptions and reduced healthcare utilization [18,19]. Previous analyses have shown that these disruptions severely affected countries with weaker health systems, exacerbating pre-existing inequities in coverage [14,20]. In addition, the incomplete recovery observed in this study suggests that pandemic-related setbacks may have had longer-lasting effects in some settings, potentially increasing the accumulation of unvaccinated and under-vaccinated children. Within this context, later-schedule vaccines, such as the second dose of measles-containing vaccine (MCV2), present the lower coverage among all the selected antigens. Unlike early-life vaccines such as DTP3 or BCG, MCV2 is administered in the second year of life, and maintaining contact with health services beyond infancy is more challenging, especially during periods of systemic disruption [21,22].

This study has several strengths, such as its global scope, extended temporal coverage, and the examination of multiple routine childhood vaccines using harmonized country-year data simultaneously. The use of multivariable models adjusted for key socioeconomic indicators strengthened the assessment of structural determinants underlying observed associations. Nevertheless, there are several limitations to consider. As an ecological analysis, the study does not allow to determine causal inference at the individual level, and associations may be influenced by contextual factors not included in the present study. In addition, the potential selection bias caused by missing data was addressed through a predefined

and transparent exclusion criterion to exclude countries without acceptable data integrity. This information bias may persist due to the heterogeneity in administrative, and survey-based coverage estimates across countries; however, this concern is mitigated by the reliance on standardized international data sources. Finally, it is important to mention that residual confounding by structural characteristics not considered in this study cannot be fully excluded.

## Conclusions

Findings from this global ecological analysis highlight that socioeconomic conditions determine patterns of routine childhood vaccination, and infant mortality at a global level. Across all vaccines analyzed, countries with lower income experienced lower vaccination coverage, and higher infant mortality, which reflected a global gradient across all the antigens assessed. While vaccination remains fundamental to child survival, the results imply that immunization efforts alone cannot offset structural disadvantages related to health system infrastructure, access to basic services, and wider social determinants. The COVID-19 pandemic further amplified these disparities by disrupting routine immunization delivery, and its recovery remained incomplete even three years later. Reductions in preventable child mortality will therefore depend on equity-oriented strategies that combine strengthened immunization with broader socioeconomic interventions.

## Supporting information

**S1 File. Includes Figs 1–6 and Tables 1-8d.**
(DOCX)

## Acknowledgments

The authors acknowledge the World Health Organization (WHO), UNICEF and the World Bank for providing the open-access data used in this study. The authors also acknowledge academic guidance provided during the development of this study.

## Author contributions

**Conceptualization:** Valeria Rios Montoya.

**Data curation:** Valeria Rios Montoya, Arlet Yelisse Suarez Alarcon, Shamira Perez Castillo, Trilce Zamudio Raez.

**Formal analysis:** Valeria Rios Montoya, Arlet Yelisse Suarez Alarcon, Shamira Perez Castillo, Trilce Zamudio Raez.

**Investigation:** Valeria Rios Montoya, Arlet Yelisse Suarez Alarcon.

**Methodology:** Valeria Rios Montoya.

**Project administration:** Valeria Rios Montoya, Antonio Marty Quispe.

**Resources:** Valeria Rios Montoya, Shamira Perez Castillo.

**Software:** Valeria Rios Montoya, Arlet Yelisse Suarez Alarcon, Shamira Perez Castillo.

**Supervision:** Antonio Marty Quispe.

**Validation:** Valeria Rios Montoya, Arlet Yelisse Suarez Alarcon.

**Visualization:** Valeria Rios Montoya, Arlet Yelisse Suarez Alarcon, Shamira Perez Castillo.

**Writing – original draft:** Valeria Rios Montoya, Antonio Marty Quispe.

**Writing – review & editing:** Valeria Rios Montoya, Arlet Yelisse Suarez Alarcon, Shamira Perez Castillo, Trilce Zamudio Raez, Antonio Marty Quispe.

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
