## [Decision Letter · Decision Letter 0]

2 Mar 2026

PONE-D-25-68661Beyond coverage: can vaccination reduce child mortality despite structural inequality? A global ecological analysis of routine childhood immunization (2010–2023)PLOS One

Dear Dr. Quispe,

Thank you for submitting your manuscript to PLOS ONE. After careful consideration, we feel that it has merit but does not fully meet PLOS ONE’s publication criteria as it currently stands. Therefore, we invite you to submit a revised version of the manuscript that addresses the points raised during the review process.

We look forward to receiving your revised manuscript.

Kind regards,

Vanya Rangelova, M.D., Ph.D.

Academic Editor

PLOS One

Journal Requirements:

[The authors have declared that no competing interests exist].

We note that one or more of the authors are employed by a commercial company: Universidad Senor de Sipan.

3. Please update your captions for your Supporting Information files at the end of your manuscript. Please renumber the captions for your Supporting Information files and update any in-text citations to ensure they follow a sequential order so it matches accordingly. Please see our Supporting Information guidelines for more information: http://journals.plos.org/plosone/s/supporting-information.

Additional Editor Comments:

on: Minor Revision

Dear Authors,

Thank you for submitting your manuscript entitled “Beyond coverage: can vaccination reduce child mortality despite structural inequality? A global ecological analysis of routine childhood immunization (2010–2023)” to PLOS ONE.

The manuscript has now been evaluated by two independent reviewers. Both reviewers find the topic timely and relevant and consider the analysis valuable for understanding the relationship between routine childhood immunization and child mortality in the context of structural inequality. They also agree that the manuscript is generally well written and methodologically sound.

However, several minor revisions are required before the manuscript can be considered for acceptance. The reviewers have provided constructive suggestions aimed at improving clarity, strengthening the presentation of the methods and results, and enhancing the interpretation of findings. Please address each reviewer comment carefully and provide a detailed point-by-point response indicating how revisions have been made.

When revising your manuscript, please ensure that:

the methodological approach and limitations are clearly described,

any language or structural issues highlighted by the reviewers are addressed.

We look forward to receiving your revised manuscript.

Sincerely,

Academic Editor

PLOS ONE

Reviewers' comments:

Reviewer's Responses to Questions

**Comments to the Author**

1. Is the manuscript technically sound, and do the data support the conclusions?

Reviewer #1: Yes

Reviewer #2: Yes

2. Has the statistical analysis been performed appropriately and rigorously?

Reviewer #1: Yes

Reviewer #2: Yes

3. Have the authors made all data underlying the findings in their manuscript fully available?

Reviewer #1: Yes

Reviewer #2: Yes

4. Is the manuscript presented in an intelligible fashion and written in standard English?

Reviewer #1: Yes

Reviewer #2: Yes

5. Review Comments to the Author

Reviewer #1: The manuscrtpts topic is highly relevant, especially given ongoing global health disparities in immunization coverage, particularly how immunization mitigates mortality in unequal settings.

The introduction effectively frames the topic's relevance, discusses equity gaps and states clear aims.

Minor suggestions:

- Delete duplicate 'for six' (line 78)

- Change “accounting country level“ to “accounting for country-level” (line 76) for grammatical accuracy

- Consider brief vaccine descriptions, noting targeted diseases, efficacy evidence, and known disparities in uptake amid structural inequalities. Listing the vaccines without context misses a chance to underscore their proven mortality reductions and equity challenges in low-resource settings

Materials and Methods are well-described and appropriate, enabling reproducibility.

Minor suggestions:

- Clarify missing data exclusion threshold

- Add statistical software and model diagnostics

- Elaborate COVID-19 disruption handling.

The results section is clearly presented, logically organized, and appropriately supported by the data, requiring no substantive revisions.

The discussion is comprehensive, well-referenced, and balanced, with excellent integration of structural determinants.

Minor suggestions:

- Change "suggest" to "suggests" (line 324) for grammatical accuracy

- Elaborate briefly on MCV2's later scheduling vulnerability

The conclusions appropriately summarize findings and offer meaningful policy implications.

Minor suggestion:

- Change “highlights” to “highlight” (line 360) for grammatical accuracy

Reviewer #2: Dear Authors,

Thank you for your work. Please find below, some minor remarks you might like to address:

1. Line 77 "for six" repeats

2. The sentence starting on line 224 "Differences across..." needs to be rewritten because currently it looks like the p-value refers to BCG vaccination coverage, while you state it is the exception.

3. Line 253, all ⍴ < 0.001 - you should substitute the rho symbol with p as you are stating a p-value.

4. Line 288 - S1 and S2 are cited in the text so they shouldn't appear in the supplementary material section. Once removed numbers should be adjusted to match numbers in the suplementary materials file.

5. The csv-file (Table S1. Country–year analytical dataset (2010–2023).) is not accessible when downloading the supplementary materials.

Kind regards

6. PLOS authors have the option to publish the peer review history of their article (what does this mean?). If published, this will include your full peer review and any attached files.

Reviewer #1: No

Reviewer #2: No

---

## [Author Response · Author response to Decision Letter 1]

26 Mar 2026

Manuscript PONE-D-25-68661

Dear Editors and Reviewers:

Thank you for the opportunity to submit a revised version of the manuscript “Beyond coverage: can vaccination reduce child mortality despite structural inequality? A global ecological analysis of routine childhood immunization (2010–2023)” for consideration in PLOS One. We are grateful for the time and effort provided, and we sincerely appreciate the feedback that has helped us to strengthen our manuscript.

We have implemented the recommendations, and the revised manuscript highlights the corresponding changes.

Reviewers' Comments to the Authors:

Reviewer 1

1. Comment from Delete duplicate “for six” (line 78)

Author response: Thank you for this observation. The duplicated phrase has been removed from the revised manuscript (line 78)

2. Comment to change “accounting country level“ to “accounting for country-level” (line 76) for grammatical accuracy

Author response: Thank you for bringing this to our attention. We have revised the phrase to “accounting for country level” in line 76.

3. Consider brief vaccine descriptions, noting targeted diseases, efficacy evidence, and known disparities in uptake amid structural inequalities.

Author response: Thank you for this valuable suggestion. A brief description of the vaccines included in the analysis, including the diseases they target and evidence supporting their effectiveness in reducing child mortality, has been added to the Data sources and variables section of the revised manuscript.

“Vaccination coverage indicators included the proportion of children receiving Bacillus Calmette-Guérin (BCG), which protects against severe forms of childhood tuberculosis; the second dose of measles-containing vaccine (MCV2), targeting measles infection and typically administered during the second year of life (around 15 to 18 months); a later schedule is associated with an increased risk of missed follow-up visits and drop-out between measles vaccine doses in routine immunization programs [21-22]; and the third dose of diphtheria-tetanus-pertussis (DTP3), hepatitis B (HepB3), Haemophilus influenzae type b (Hib3), and poliovirus (Pol3) vaccines, which prevent major infectious diseases such as diphtheria, tetanus, pertussis, hepatitis B infection, invasive Haemophilus influenzae type b disease, and poliomyelitis, respectively. These antigens are the core components of routine child immunization programs and have contributed substantially to global reduction in vaccine-preventable child mortality. Despite this effectiveness, vaccination coverage remains uneven across countries, reflecting structural inequalities, including socioeconomic disadvantage, geographical barriers to healthcare access, and variations in health system capacity, particularly in low-income settings.”

4. Comment from Minor suggestions:

- Clarify missing data exclusion threshold

- Add statistical software and model diagnostics

- Elaborate COVID-19 disruption handling.

Author response: Thank you for these beneficial suggestions. We have addressed these points in the revised manuscript, specifically in the Materials and Methods section.

“To ensure data quality and comparability, the integrity of country-year observations was evaluated across all variables; the number of missing values (NAs) per country ranged from 84 to 540 across the study period. Countries with more than one-third of the maximum observed number of missing values (>180 NAs) were excluded from the analysis. This threshold was applied to minimize bias from excessive missing data while the final analytical sample was restricted to countries meeting this inclusion criterion.”

“ In addition, because routine immunization services were widely disrupted during the COVID-19 pandemic, the study period (2010-2023) was intentionally retained to capture both pre-pandemic trends and pandemic-related deviations in vaccine coverage. Rather than excluding these observations, country-year data from 2020 to 2023 were maintained to evaluate changes in coverage trajectories. Temporal trends were, therefore, interpreted considering this global disruption, which allowed the analysis to capture both the abrupt decline in vaccination coverage in 2020 and the subsequent partial recovery during 2021-2023.”

“All statistical analyses were conducted using R version 2024.04.2. Model assumptions were evaluated through standard diagnostic procedures, including inspection of residual distributions and assessment of multicollinearity among predictors.”

5. Line 324: Change “suggest” to “suggests.”

Author response: Thank you for pointing this out. The correction has been implemented in the revised version of the manuscript.

6. Elaborate briefly on MCV2’s later scheduling vulnerability.

Author response: Thank you for this suggestion. A brief explanation regarding the later scheduling of MCV2 and its potential vulnerability to missed vaccinations or service disruptions has been added to the Data sources and variables section of the revised manuscript.

“ …the second dose of measles-containing vaccine (MCV2) targets measles infection and is typically administered during the second year of life (around 15 to 18 months); a later schedule is associated with an increased risk of missed follow-up visits and drop-out between measles vaccine doses in routine immunization programs [21-22].”

7. Line 360: Change “highlights” to “highlight.”

Author response: Thank you for pointing this out. The correction has already been made in the revised version of the manuscript.

Reviewer 2

Dear Authors,

Thank you for your work. Please find below, some minor remarks you might like to address:

1. Line 77 "for six" repeats

Author response: Thank you for pointing this out. The correction has already been made in the revised version of the manuscript.

2. The sentence starting on line 224 "Differences across..." needs to be rewritten because currently it looks like the p-value refers to BCG vaccination coverage, while you state it is the exception.

Author response: We agree with the reviewer’s suggestion. The correction has already been made to make it clearer that we wanted to state it was the exception.

3. Line 253, all ⍴ < 0.001 - you should substitute the rho symbol with p as you are stating a p-value.

Author response: Thank you for pointing this out; the rho symbol has already been replaced with “p”.

4. Line 288 - S1 and S2 are cited in the text so they shouldn't appear in the supplementary material section. Once removed numbers should be adjusted to match numbers in the supplementary materials file.

Author response: The reviewer does not clarify if they are referring to S1/S2 Table or S1/S2 Figure; however, none of them are cited in the text, especially not in line 288, so we are not sure of what the reviewer is referring to.

5. The csv-file (Table S1. Country–year analytical dataset (2010–2023). is not accessible when downloading the supplementary materials.

Author response: The table the author is referring to is Table S3 in the CSV file. The code writes two CSV files that can be opened in Excel so that they are understandable. The code writes two CSV files that can be opened in Excel in order for it to be understandable.

---

## [Editor Report · Decision Letter 1]

21 Apr 2026

Beyond coverage: can vaccination reduce child mortality despite structural inequality? A global ecological analysis of routine childhood immunization (2010–2023)

PONE-D-25-68661R1

Dear Antonio Marty Quispe

We’re pleased to inform you that your manuscript has been judged scientifically suitable for publication and will be formally accepted for publication once it meets all outstanding technical requirements.

Kind regards,

Vanya Rangelova, M.D., Ph.D.

Academic Editor

PLOS One

Additional Editor Comments (optional):

Dear Authors,

Thank you for submitting your manuscript entitled “Beyond coverage: can vaccination reduce child mortality despite structural inequality? A global ecological analysis of routine childhood immunization (2010–2023)” to PLOS ONE.

I have now carefully considered the revised version of your manuscript, the reviewers’ comments, and your detailed responses to the points raised during peer review. I am pleased to inform you that your manuscript has been accepted for publication in PLOS ONE.

Both reviewers found the study to be timely, relevant, and methodologically sound, and they considered the revisions to have adequately addressed the minor concerns raised in the initial review round. Your responses were clear and constructive, and the revised manuscript has been strengthened accordingly.

This study makes a valuable contribution to the literature by examining the relationship between routine childhood immunization coverage, structural inequality, and child mortality at a global scale over an extended time period. The findings provide important evidence for researchers, policymakers, and public health practitioners seeking to understand how vaccination programs may contribute to improved child survival outcomes even in settings marked by persistent socioeconomic disparities.

On behalf of the journal, I thank you for choosing PLOS ONE for the dissemination of your work. We appreciate your contribution and look forward to seeing your article published.

With best regards,

Vanya Rangelova

Academic Editor
---

## [Editor Report · Acceptance letter]

PONE-D-25-68661R1

PLOS One

Dear Dr. Quispe,

I'm pleased to inform you that your manuscript has been deemed suitable for publication in PLOS One. Congratulations! Your manuscript is now being handed over to our production team.

Kind regards,

on behalf of

Dr. Vanya Rangelova

Academic Editor

PLOS One